Swarm intelligence-based packet scheduling for future intelligent networks

Husen Arif arifhrashid@gmail.com 1 2
Hasanain Chaudary Muhammad 1
Ahmad Farooq 1
Farooq-i-Azam Muhammad 3
Hwang See Chan 4
Ghani Arfan 5
1 Department of Computer Science, COMSATS University Islamabad , Lahore , Punjab , Pakistan
2 Department of Computer Science and Information Technology, Virtual University of Pakistan , Lahore , Punjab , Pakistan
3 Department of Electrical and Computer Engineering, COMSATS University Islamabad , Lahore , Punjab , Pakistan
4 School of Computing, Engineering and the Built Environment, Edinburgh Napier University , Edinburgh , Scotland , United Kingdom
5 Department of Computer Science and Engineering, School of Engineering, American University of Ras al Khaimah , Ras al Khaimah , United Arab Emirates
Jhanjhi N
Electronic publication date: 2023 Nov 16
Publication date: 2023
Volume: 9
Electronic Location ID: e1671
Received 2023 Apr 14; Accepted 2023 Oct 8
Copyright: ©2023 Husen et al.
Copyright year: 2023
Copyright holder: Husen et al.
License: This is an open access article distributed under the terms of the Creative Commons Attribution License, which permits unrestricted use, distribution, reproduction and adaptation in any medium and for any purpose provided that it is properly attributed. For attribution, the original author(s), title, publication source (PeerJ Computer Science) and either DOI or URL of the article must be cited.
License URL: https://creativecommons.org/licenses/by/4.0/

Keywords: TIPS, Machine learning, Data mining, Emerging technologies

Funding: There is no funding involved in this research.

==============================
Network operations involve several decision-making tasks. Some of these tasks are related to operators, such as extending the footprint or upgrading the network capacity. Other decision tasks are related to network functions, such as traffic classifications, scheduling, capacity, coverage trade-offs, and policy enforcement. These decisions are often decentralized, and each network node makes its own decisions based on the preconfigured rules or policies. To ensure effectiveness, it is essential that planning and functional decisions are in harmony. However, human intervention-based decisions are subject to high costs, delays, and mistakes. On the other hand, machine learning has been used in different fields of life to automate decision processes intelligently. Similarly, future intelligent networks are also expected to see an intense use of machine learning and artificial intelligence techniques for functional and operational automation. This article investigates the current state-of-the-art methods for packet scheduling and related decision processes. Furthermore, it proposes a machine learning-based approach for packet scheduling for agile and cost-effective networks to address various issues and challenges. The analysis of the experimental results shows that the proposed deep learning-based approach can successfully address the challenges without compromising the network performance. For example, it has been seen that with mean absolute error from 6.38 to 8.41 using the proposed deep learning model, the packet scheduling can maintain 99.95% throughput, 99.97% delay, and 99.94% jitter, which are much better as compared to the statically configured traffic profiles.

Introduction

Different network resources such as bandwidth, power, and spectrum are never abundant, regardless of the significant developments in the network technologies, to provide higher capacities and minimize delays (Huang et al., 2023; OECD, 2022; Shen & Chen, 2018; Sheng et al., 2019). On the other hand, the traffic growth is immense due to rising trends in digitization and network-enabled devices from the Internet of Vehicles (IoV), Tactile Internet (TI), and Internet of Things (IoT) (Cisco, 2021). Moreover, the bandwidth requirements from the standard services are also rising due to increased quality and data and latency constraints. As a result, network operators are often challenged to implement suitable oversubscription models (Ni, Huang & Wu, 2014) and differentiated service bundling mechanisms that allow them to ensure Return on Investment (ROI). For example, packet scheduling (Beams, Kannan & Angel, 2021) is a common approach to controlling service quality in oversubscribed networks where the traffic of different types needs to be handled according to their service package or the contracted service level agreements (SLA).

Network deployments often follow a comprehensive planning process that considers several demographic and statistical factors in addition to seasonal and regular network user behaviors. The planning processes often utilize the type of services, the technologies, the required volume of network resources, and the target SLA (Gavriluţ, Pruski & Berger, 2022). Traditionally, for cellular networks, the erlang traffic intensities describe the expected load that networks must withstand in different types of points of presence (POP) (Głkabowski, Hanczewski & Stasiak, 2015). The loads correspond to business models, target ROI, and other policies of the network operators.

Traditional packet scheduling techniques utilize several factors, including non-differentiated approaches where traffic scheduling takes place without differences, such as First-in-First-out (FIFO), Fair Queueing (FQ), Priority Queuing (PQ), and Round Robin (RR) (Medhi and Ramasamy, 2018). Differentiated approaches, on the other hand, include classifying traffic into distinct classes and assigning weights or priorities to the classes. The latter approaches often employ non-differentiated approaches within a class or priority group. Recently, it has been studied that significant Quality of Service (QoS) and network utilization improvements can be made with packet scheduling approaches that consider features, such as time and origin characteristics (TAOC) of traffic (Husen et al., 2021; Rashid & Muhammad, 2019). The TAOC characterization is related to the network planning processes, such as traffic forecasting and its breakdown to the level of traffic origins, formally known as POPs. The TAOC characterization concentrates on novel features, such as Origin Class Feature (OCF), Volume Feature (VF), Time Feature (TF), Traffic Intensity Feature (TIF), and Network Resource Feature (NRF).

This article investigates deep learning (DL) based techniques for learning the TAOC-based packet scheduling algorithms. Following are the contributions of this research,

1. The existing packet scheduling approaches are analyzed in the context of intelligent methods.

2. A novel DL-based traffic profiling scheme is proposed to fulfill the swarm intelligence requirement of traffic intensity-based packet scheduling.

3. The traffic intensity-based packet scheduling algorithm is ported to Network Simulator 3 (NS3) for experimental evaluations and analysis.

4. An analysis of the DL-based profiling scheme and network performance objectives is presented.

Contributions (1) and (2) are required for implementing TAOC-aware packet scheduling to bridge the gap between the network planning processes and the functional network decision-making, such as packet scheduling. The framework also considers network node role awareness while making packet scheduling decisions.

The rest of the article is divided into the sections as follows: Materials & Methods, Results, Discussion, and Conclusion. First, the Material and Methods section covers the packet scheduling strategies, learning mechanism for TAOC, high-level framework for TAOC-based packet scheduling, DL-based model, and its integration with the TAOC-based packet scheduling algorithm. Next, the Results section presents the experimental setup and evaluation metrics for the DL model and network performance. The Discussion section focuses on the significance of the results and future research directions. The concluding remarks are provided in the Conclusions section.

Materials & Methods

Packet scheduling approaches

There are a large number of packet scheduling techniques that work based on traditional factors, such as weight, priority, class, and arrival order. However, there are very few techniques proposed that consider TAOC characteristics. A prominent scheme in this regard has been proposed by Husen et al. (2021) and Rashid & Muhammad (2019) that considers the TAOC characteristics and network node and layer role. However, the study does not consider the effects of any machine learning-based real-time TAOC characterization scheme.

The traditional packet scheduling schemes can be categorized into reactive and proactive types. The reactive schemes monitor specific parameters, and on a change, the scheduling decisions are updated, such as in relatively differentiated scheduling (Striegel & Manimaran, 2002), preemption-based scheduling (Miao et al., 2015), and queue length-based delay-aware packet scheduler (Yu, Znati & Yang, 2015). In addition, the active time fair queuing (Zhang et al., 2015), competitive rate-based scheduling (Deshmukh & Vaze, 2016), dynamically weighted low complexity fair queuing (Patel & Dalal, 2016), time-frequency resource conversion (Sungjoo et al., 2016), modified first come first serve (Xu et al., 2016) and efficient and flexible software packet scheduling (Saeed et al., 2019) are also reactive schemes proposed in the recent literature. On the other hand, the proactive schemes adjust the decision parameters before the changes occur, such as in the multiple dimensions of locality-based scheduling (Iqbal et al., 2016), multi-generation packet scheduling (Huang, Izquierdo & Hao, 2017), D2-Pas (Zhang et al., 2019a; Zhang et al., 2019b), calendar queuing (Sharma et al., 2020) and traffic intensity-based packet scheduling (TIPS) (Husen et al., 2021; Rashid & Muhammad, 2019).

Table 1 Comparison of existing packet scheduling techniques.

Ref	Year	Scheme	Approach	TAOC Features	
				VF	OCF	AL	OC	TF	TIF	NRF	
Striegel & Manimaran (2002)	2002	Relatively differentiated scheduling	Delay, loss, throughput	✓	–	–	–	–	–	–	
Miao et al. (2015)	2015	Preemption-based packet-scheduling scheme	Fairness Based	–	–	–	–	–	–	–	
Iqbal et al. (2016)	2015	Multiple dimensions of locality	Minimize the out-of-order packets	–	✓	–	–	–	–	–	
Yu, Znati & Yang (2015)	2015	Queue length-based delay-aware packet scheduler	congestion- and energy-aware packet scheduling scheme	–	–	–	–	–	–	–	
Han et al. (2015)	2015	stochastic packet scheduling		–	–	–	–	–	–	–	
Lee & Choi (2015)	2015	Group-based multi-level packet scheduling	Group-based multilevel packet scheduling	–	–	–	–	–	–	–	
Zhang et al. (2015)	2015	Active time fairness queuing	Active time fairness queuing	–	–	–	–	–	–	–	
Deshmukh & Vaze (2016)	2016	Competitive rate based	Packet arrival	✓	–	–	–	–	–	–	
Patel & Dalal (2016)	2016	Dynamically weighted low complexity fair queuing	Weighted fair queuing	–	–	–	–	–	–	–	
Sungjoo et al. (2016)	2016	Time-frequency resource conversion	Available bandwidth constraints	–	–	–	–	–	–	–	
Xu et al. (2016)	2016	Modified first come first serve	Delay constraints	–	–	–	–	–	–	–	
Pavithira & Prabakaran (2016)	2016	Binary search algorithm	Binary search tree	–	–	–	–	–	–	–	
Huang, Izquierdo & Hao (2017)	2017	Multi-generation packet scheduling	Redundant packets	–	–	–	–	–	–	–	
Zhang et al. (2019a) and Zhang et al. (2019b)	2019	Distributed dynamic packet scheduling	constructs schedule locally at individual nodes	–	–	–	✓	–	–	–	
Saeed et al. (2019)	2019	Efficient and flexible packet scheduling	Packet ranking, Find First Set (FFS), priority	–	–	–	–	–	–	–	
Karimi et al. (2019)	2019	Channel state and user requirements	Latency, control channel, HARQ, and radio channel aware	–	–	–	–	–	–	–	
Sharma et al. (2020)	2020	Calendar queue	Prioritization dynamic escalation of packet priorities	–	–	–	–	✓	–	–	
Wei et al. (2020)	2020	Shared bottleneck-based congestion control scheme	Shared bottlenecks among sub-flows and estimated the congestion degree of each sub-flow.	–	–	–	–	–	–	–	
Husen et al. (2021)	2021	Traffic intensity-based packet scheduling	Traffic intensity variations	✓	✓	–	✓	✓	✓	✓	
Yu et al. (2021)	2021	Admission-In First-Out	Maintain a sliding window to track the ranks	–	–	–	–	–	–	–	

A summary of a comparison of existing techniques is given in Table 1. The comparison focuses on the TAOC features, namely the Volume Feature (VF), OCF, Automated Learning (AL), OC, Time Feature (TF), Traffic Intensity Feature (TIF), and Network Resource Feature (NRF). Most existing studies try to address one or more issues related to network performance and do not consider the requirements of automated learning for future intelligent networks. The relatively differentiated scheme proposed by Striegel & Manimaran (2002) uses the traffic volume to improve the throughput and minimize the delay and packet loss. It does not consider the current state of network resources regarding NRF, TIF, OC, and TF and lacks the essential AL capability.

Another approach based on the multiple dimensions of locality, proposed by Iqbal et al. (2016) explicitly uses the OCF to minimize the out-of-order packets (OOP). The minimization of OOP can reduce buffer overflows on the receiver side, providing an advantage if the performance is maintained. However, it does not consider the effects of VF, TF, TIF, and NRF and does not support the AL.

Another scheme that considers the VF has been proposed by Deshmukh & Vaze (2016), which measures the packet arrival rates and establishes their competitive ratios to schedule the packets. In another effort, Zhang et al. (2019a) and Zhang et al. (2019b) studied a distributed dynamic packet scheduling scheme that uses the OC to schedule the packets on local nodes. Similarly, Sharma et al. (2020) used the TF for their calendar queuing-based packet scheduler. It dynamically adjusts the escalation of the packet priorities using the TF. Although the three schemes above partially consider the TAOC features, none consider the AL and NRF.

The scheme proposed by Husen et al. (2021) uses most of the features of TAOC, namely the VF, OCF, OC, TF, TIF, and NRF. The authors have presented extensive experiments to show that the scheme effectively improves resource utilization and traffic performance, especially in constrained networks. However, the scheme does not support AL and relies on domain expertise and network dimensioning processes. This research will exploit the scheme above and enable the AL to ensure network utilization and performance with agility and cost-effectiveness.

Some other common techniques proposed by researchers do not support several features of the TAOC. The details of the schemes have been compared and summarized in Table 1.

It can be observed from the above analysis that few techniques consider the TAOC features, which are partial except for the TIPS scheme. The TIPS technique was implemented and evaluated with predefined TAOC features and did not learn from actual live streams of the traffic. Using predefined TAOC features is a tedious and complex process, as it may take extensive time to conclude the features and requires domain knowledge and human intervention.

Learning TAOC features

Learning the TAOC characteristics involves classifying traffic according to volume, origin, and periodic variations, as shown in Fig. 1. In addition, suitable techniques are required to approximate the historical TOAC features and use these features to predict the traffic intensity at a given or future point in time.

Figure 1 Components of swarm intelligence for packet scheduling and TAOC characterization.

In the recent era, future intelligent networks (FIN) (Husen, Chaudary & Ahmad, 2022) have been envisioned, which are expected to use ML techniques to automate computation, network functions, and operations intelligently. For example, the ML-based intelligent packet scheduling techniques may learn the network state and usage behavior and control the scheduling decisions on different types of network nodes. The ML network techniques provide autonomous, cheaper, and faster decision-making in managing network functionalities.

Recent literature has several studies on ML-based techniques for traffic classification and forecasting. The ML models used in these studies classify the traffic streams. In addition, they forecast their behavior in the future time based on learning the insights from the historical usage of the data. The ML models have been used to extract spatiotemporal features, some of which are also part of TAOC characterization. However, they have not yet been considered for packet scheduling decisions. For TAOC, specific ML models are required for traffic classification based on their origin. This classification differs from the traditional classification used for packet scheduling, which is based on the type of flow, source and destination addresses, port numbers, or applications. For the origin-based traffic classification, the classes represent the different groups based on the origin areas, such as serving areas or POPs. The time feature represents the traffic variations per OCF and captures how traffic patterns change with time, day, week, month, year, or season. The Volume Feature (VF) captures the variations of the traffic volumes in terms of the number of packets. It also needs to cover the packet size variations, thus covering the maximum capacity the switching system has to handle.

Traffic intensity features involve deriving the erlang values for the given OC and VF. The features mentioned above are used for OC trend curves, and packet schedulers generate the queues according to the number of OCs. Packets are processed from the input interfaces to output interfaces in precedence of the predicted values. It can be noticed from Table 2 that the existing works for traffic classification and prediction do not cover all the features required for TAOC characterization. However, the studies consider several partial requirements, such as spatiotemporal features addressed by Zhang et al. (2019a), Zhang et al. (2019b), He, Chow & Zhang (2019), Bega et al. (2019), Wang et al. (2017), and Wass (2021). However, these studies do not cover the entire scope of traffic forecasting. For example, they cannot generate the classes of traffic based on OC and their approximations using time, volume, and traffic intensity.

Table 2 Summary of traffic prediction strategies.

Ref	ML Model	Scope	TAOC Features	
			Type	OC	TF	VF	TIF	
Nguyen et al. (2012)	Naïve Bayes and C4.5 Classifier	Classification of Interactive IP traffic	Y	X	X	X	X	
Singh & Agrawal (2011)	MLP, RBF, C4.5, Bayes Net and Naïve Bayes	IP Traffic Classifications	Y	X	X	X	X	
Zhang et al. (2019a) and Zhang et al. (2019b)	Spatial-Temporal Cross-domain Neural Network	Traffic Forecasting	Y	Y	Y	X	X	
He, Chow & Zhang (2019)	Spatio-Temporal Convolutional Neural Network	Traffic Forecasting	Y	X	Y	X	X	
Reddy & Hota (2013)	Naïve Bayes classifier, Bayesian Network, Decision trees, and Stacking and Voting	P2P traffic	Y	X	X	X	X	
Bega et al. (2019)	DeepCog	Capacity Forecasting	Y	X	Y	Y	X	
Wang et al. (2017)	Autoencoder and LSTM	Mobile Traffic Forecasting	X	Y	Y	X	X	
Wass (2021)	Attention Transformer model	Mobile Traffic Prediction	X	Y	Y	Y	X	

Similar is the case with traffic classification, addressed in several existing studies such as Nguyen et al. (2012), Singh & Agrawal (2011), Zhang et al. (2019a), Zhang et al. (2019b) and Reddy & Hota (2013). The scope of these studies is very limited to identifying one class versus other specific application traffic. Therefore, further work is required to investigate traffic classification and forecasting based on the TAOC features.

High-level framework for TAOC-based packet scheduling

This section focuses on a high-level framework for ML-based TAOC packet scheduling, which covers several important aspects of packet scheduling. The benefits of the TAOC features include user traffic classification, prediction, and network state regarding bandwidth capacity. The high-level implementation of the ML-based TAOC packet scheduling is shown in Fig. 2. It employs several network layer-specific learners, and the learning process and forecasting process are distributed.

Figure 2 A swarm intelligence-based TAOC packet scheduling.

The Base Learners (BL) provide the OC classification and approximation of the TF, VF, and TIF features. The predictions from BLs are used to manipulate scheduling on perimeters of networks, such as the access or backhaul nodes.

Distribution-Aggregate Learners (DAL) aggregate the learnings from the BLs. The DAL does not learn from the actual traffic on distribution or aggregation nodes; rather, it employs the learnings from the BLs and provides the necessary information for deciding on distribution nodes. The DALs are node role-specific learners and incorporate the effects of oversubscription.

Similarly, the Core Aggregate Leaner (CAL) and Edge Aggregate Learners (EAL) aggregate the learning from the DAL and incorporate the node role and oversubscription effects. The EALs are based on CAL learnings and incorporate the breakout ratio of the traffic, i.e., the traffic that will leave the network perimeters.

The predictions from each of the learners are provided as input to the respective TIPS schedulers (Rashid & Muhammad, 2019), such as Access-Packet Scheduler (APS), Distribution-Packet Scheduler (DPS), Core-Packet Scheduler (CPS), and Edge-Packet Schedulers (EPS).

TAOC learning with LSTM

Erlang distribution can represent the TAOC characteristics. On the other hand, recurrent neural networks (RNN) (Jain & Medsker, 1999), as shown in Fig. 3, is a deep learning paradigm suitable for predicting the Erlang distribution based on the traffic characteristics of each access node. Moreover, since the traffic originating from a given node depends on the previous pattern, the patterns are generally interlinked and may also repeat in future time intervals. Finally, the traffic patterns depend on the serving area’s demographics. Traditional neural networks can be used to characterize traffic with temporal dependencies; however, RNNs, due to their specific architecture, can address the issues in a better manner.

Figure 3 Recurrent neural network.

An analysis of LSTM

Long short-term memory (LSTM) (Houdt, Mosquera & Nápoles, 2020) is a special RNN cell that addresses the issues faced with traditional RNNs, such as loss of the long-term dependencies due to vanishing gradient issues, as discussed in Bengio, Simard & Frasconi (1994).

Figure 4 shows that an LSTM employs three gates to control adding or removing information from the cell state. The forget gate (ft) defines what information needs to be removed from the cell state. It analyses the previous state (yp) and current input sequence (xt).

Figure 4 LSTM Structure for TAOC.

The ft is defined in Eq. (1), and its output is always between 0 and 1 for each state in the cp where zero means discarding it. (1) ft=σWf.yp,xt+bf

The information that needs to be retained in the current state of the cell is determined in two steps. First, the input gate (it) function, as defined in Eq. (2), determines the parameters that need to be updated. Then, the candidate parameter values (c ~t) are generated by the tanh function as per Eq. (3). Next, these are combined to update the values of ct as per Eq. (4).

(2) it=σWf.yp,xt+bi

(3) c ~t= tanhWc.yp,xt+bc

(4) ct=ft ∗cp+it ∗c ~t

The output of the LSTM is based on the current state, and the sigmoid function is applied to select the values to be output. The final output of the cell is obtained by applying the tanh function to the above values and multiplied by the output gate. The output gate function is defined in Eq. (5), and the current output state is defined in Eq. (6).

(5) ot=σWo.yp,xt+bo

(6) yt=ot ∗tanhct

TAOC learning model (TLM)

TIPS scheduling algorithm requires five different values, including the time, node function, number of downstream nodes, and rank and weight of the erlang values. Therefore, the objective of the TLM is to characterize each access node and provide the above mentioned information to the upstream node. To cope with the above requirements, an LSTM-based encoder–decoder model is adopted, as shown in Fig. 5.

Figure 5 TLM unit model.

The above model is replicated for each physical interface on the downstream side. In addition to the unit model for each interface, an aggregate model combines the input and output values. The model takes the ts number of sample values of x and predicts the m number of values for t + 1, T + 2, …, t + m.

In the unit model, Cei represents the LSTM cells for the encoder where i falls in t,t+n and Cdj represents the LSTM cells for the decoder part. The j ranges from t + 1 to t + m. The DLd forms the dense layer, which receives the state information from all the cells and transforms it into the output of Ld dimension. On the other hand, D1 is a dense layer (Helen Josephine, Nirmala & Alluri, 2021) that receives the m inputs and connects the F layer to connect a single dimension stream. The F flattening layer (Chen et al., 2023) that receives the inputs equals the hidden layers ( Le) multiplied by the ts units.

The TLM is installed on each network node with the downstream nodes, such as the aggregation layer (DAL) and core nodes (CAL). The core nodes receive the swarm of g inputs fed to TLM(c) nodes. The TLM(g) nodes provide the data to TIPS instances running on G nodes, and TLM(c) provides the data for the TIPS instances running on C nodes, as shown in Fig. 6.

Figure 6 TLM on downstream and upstream nodes.

There are two approaches available for implementing TLM on the DAL layer. The first approach is to install TLM on the egress interface of the access nodes, which predicts the dim+1 used by the TLM instances running on DAL. This approach is efficient for large-scale networks and allows the distributed learning model. The other approach is to implement TLM on the ingress interfaces of aggregation nodes, i.e., DAL nodes. In this case, it serves two purposes; the TLM provides the base learning as well as dim+1 in addition to gim+1.

Results

This section covers the details of the experimental setup, various parameters, evaluation metrics, and their measured results. First, mean absolute error (MAE) and mean squared error (MSE) are used for the TLM evaluation to determine the difference from the actual traffic features. Next, the throughput, delay, and jitter values are measured for the network performance. Finally, the cumulative distribution function (CDF) is used to depict the results of different measurements, as it can show how the different metrics vary throughout the experiment.

Experimental setup

The TLM was implemented using the KERAS library (Gulli & Pal, 2017) in Python for experimental analysis, and the source code is available as supplementary material. The TIPS was originally implemented for NS2 and was ported to Network Simulator 3 (NS3) (Riley & Henderson, 2010) to integrate the TLM.

Network topology

Figure 7 shows the experimental network setup following the state-of-the-art hierarchical topology. The hierarchal network topologies offer several benefits in scalability, redundancy, performance, security, manageability, and maintenance (Cisco, 2022; Zhang & Liang, 2008). The access nodes represent a serving area with a distinct traffic profile. The access nodes to aggregation nodes connectivity is achieved with dedicated physical links with different data rates according to the requirements of the serving area. Furthermore, the access node to aggregation nodes connectivity follows the rule of proximity, i.e., the access nodes from co-locating areas are connected to the same aggregation node.

Figure 7 Experimental network topology.

For the experimental setup, traffic sources are configured to generate the traffic with Gaussian distribution, a well-known internet traffic distribution discussed in Bothe, Qureshi & Imran (2019) and Meent, Mandjes & Pras (2006). Although all the access nodes will follow a Gaussian distribution, their mean values and shapes are different for different types of demographics of the serving area, as discussed in a study conducted by Husen et al. (2021).

The specifications of different links are given in Table 3. The access to aggregation links follows random allocation for each experiment instance and uses a FIFO scheduler. The TIPS is not used on access links as it represents a single traffic profile. On access to aggregation nodes, the link’s capacity is set to 4 Mbps, and aggregation to the core link has a 6 Mbps capacity. Both of the above links use the TIPS packet scheduler. The edge links connecting the destination nodes use the FIFO scheduler as there is only a single type of traffic profile.

Table 3 Specifications of different links.

Link	Capacity	Delay	Packet Scheduler	
Ba1, Ba2, Ba3	[500 Kbps, 750 Kbps, 1 Mbps, 1.25 Mbps]	[2 ms, 2 ms, 2 ms, 2 ms]	FIFO	
Bg1, Bg2,Bg3	4 Mbps	2 ms	TIPS	
Bc	6 Mbps	2 ms	TIPS	
Be	5 Kbps	2 ms	FIFO	

Traffic Intensity based Packet Scheduling

The Traffic Intensity based Packet Scheduling (TIPS) is a packet scheduler that schedules packets according to the traffic profile of the downstream links. Packets are enqueued to different queues with dynamic dequeue order following the ranks of the traffic profile, and the size of the dequeue is dependent on the weights of traffic profiles. The control traffic is separated and handled with different policies different from TIPS. The TIPS was initially developed for NS2 (Husen et al., 2021); however, since the NS2 development has stopped, it was ported to Network Simulator 3 (NS3) (Riley & Henderson, 2010). The implementation of TIPS for NS3 is available in the supplementary material of this paper.

TLM performance results and analysis

This section focuses on analyzing the traffic generated from the serving areas, evaluating the performance of TLM on aggregation and core nodes, and then, the network performance analysis, which includes throughput, delay, and jitter. Finally, the network performance shall be compared with the results obtained without TLM.

Access traffic profiles

The TLM performance’s primary expectation is traffic profile diversity. Several experiments were conducted with different traffic profiles. Distinct traffic profiles were generated for each experiment iteration with separate distribution parameters such as mean and standard deviation. A set of randomly selected traffic profiles generated according to Gaussian distribution are shown in Figs. 8, 9 and 10 for different aggregation nodes to demonstrate the effects of Gaussian parameters. The lower and upper parts of the graphs of the figures above show the cumulative density and histogram of the data rates generated by the respective node. The histogram on the top part of the graphs shows that an access node’s traffic follows a Gaussian distribution (Yamanaka & Usuba, 2020). This implies that for a given mean value, there are equal chances of traffic generated below or above.

Figure 8 Traffic measured at aggregation node 1.

Figure 9 Traffic measured at aggregation node 2.

Figure 10 Traffic measured at aggregation node 3.

Prediction accuracy of TLMs

This section analyses the performance of the TLM models on different nodes. The TLM learners are used on G1, G2, G3, and C1 as per the selected links. The prediction accuracy of different TLMs is shown in Figs. 11, 12, 13 and 14. The figures present the histograms of the actual data rates and those predicted by the respective TLM. A summary of the mean absolute error (MAE) and mean squared error (MSE) is given in Table 4, which shows the MAE values between 8.41 to 6.38 for TLM on nodes G1, G2, G3, and C1. Since the TLMs coordinate with each other to realize swarm learning, the latency of the inter TLMs may pose a limitation. In practice, to address it, low-latency dedicated communication links can be used to address it. The latency issues between inter-TLM communications can also be minimized by implementing the TLMs on the same location as the nodes of the network. The location of TLMs for different network nodes is an essential factor to ensure optimal performance. In this work, the TLMs were implemented in the same location as the respective network nodes to eliminate the effects of the latency. If the TLMs are implemented in a centralized location to avail benefits of the cloud computing paradigm, low latency dedicated networks would be required.

Figure 11 The TLM prediction accuracy on the aggregate node (G1).

(A) The figure shows the histogram of aggregate TLM (G1) prediction accuracy. (B) The brown color represents the TLM’s predicted data rate values. (C) The black color represents the actual data rate values.

Figure 12 The TLM prediction accuracy on the aggregate node (G2).

(A) The figure shows the histogram of aggregate TLM (G2) prediction accuracy. (B) The brown color represents the TLM’s predicted data rate values. (C) The black color represents the actual data rate values.

Figure 13 The TLM prediction accuracy on the aggregate node (G3).

(A) The figure shows the histogram of aggregate TLM (G3) prediction accuracy. (B) The brown color represents the TLM’s predicted data rate values(C) The black color represents the actual data rate values.

Figure 14 The TLM prediction accuracy on the core node (C1).

(A) The figure shows the histogram of core TLM prediction accuracy. (B) The brown color represents the TLM’s predicted data rate values (C) The black color represents the actual data rate values.

Table 4 Summary of performance of TLM.

Node	MAE	MSE	
G1	8.41	72.83	
G2	7.19	53.25	
G3	6.38	42.68	
C1	7.33	54.22	

Network performance analysis of TLM

This section evaluates the network performance using TIPS with TLM as a packet scheduler on links on aggregation and core nodes. The throughput, source-to-destination delay, and jitter are measured with and without TLM.

Figure 15 shows the maximum throughput achieved with and without TLM. Without TLM, TIPS uses statically configured profiles. The TLM learns the traffic profiles of access nodes on run time and provides necessary information for scheduling decisions. It can be seen that the TLM achieves the same throughput as the manually engineered profiles. For both cases, the throughput achieved was 2,000 Kbps for almost 50% of measurements. The average difference with the TLM algorithm I observed to be 0.05% in throughput.

Figure 15 Comparison of maximum throughput (Kbps).

(A) The upper part of the figure shows the throughput histogram curves with and without TLM. (B) The lower part of the figure shows the CDF curves with and without TLM. (C) The gold color refers to the throughput values with TLM and black color refers to the throughput values without TLM.

Similarly, Fig. 16 shows the maximum delay achieved with and without TLM. Without TLM, TIPS uses statically configured profiles. It can be seen that delay measurements with TLM are the same as with the manually engineered profiles. For both cases, the maximum delay is centered around 15 to 17.5 ms. The average delay difference with TLM is observed to be 0.03%.

Figure 16 Comparison of maximum delay from source to destination (milliseconds).

(A) The upper part of the figure shows the delay histogram curves with and without TLM. (B) The lower part of the figure shows the CDF curves with and without TLM. (C) The gold color refers to the delay values with TLM and the black color refers to the delay values without TLM.

Finally, Fig. 17 shows the maximum jitter achieved with and without TLM. Without TLM, TIPS uses statically configured profiles. Similar to delay results, it can be seen that the TLM maintains the same jitter as the manually engineered profiles. The maximum jitter is around two milliseconds for both cases. The average jitter difference with TLM is 0.05%, which shows that it can limit the jitter by 99.95%, as achieved with the manual configurations.

Figure 17 Comparison of maximum jitter (milliseconds).

(A) The upper part of the figure shows the jitter histogram curves with and without TLM. (B) The lower part of the figure shows the CDF curves with and without TLM. (C) The gold color refers to the jitter values with TLM and black color refers to the jitter values without TLM.

Discussion

The TIPS has been compared with several state-of-the-art scheduling algorithms (Husen et al., 2021), and it has shown that TIPS provides advantages in terms of throughput, delay, and jitter in congested networks. This research has evaluated the automated traffic profile learning with the deep learning model, TLM, and compared it with traffic profiles statically configured on the network nodes through the network dimensioning processes. The efficiency of TLM in learning the TAOC characteristics is demonstrated and shows that the performance is better than the statically configured profiles.

The intelligent automation of future network functions and processes is essential. It has been envisioned in several recent articles such as Brito, Mendes & Gontijo (2020), Wang et al. (2020), and Zhu et al. (2020), where ML-based automation has been indicated as the primary requirement. Several researchers have already started investigations to integrate the intelligence into the network functionalities, such as the Kalman filter for predictive resource allocation (Teixeira & Timóteo, 2021) and model-free Kalman-Takens filter for the signal-to-noise ratio prediction in 5G networks (Teixeira & Timóteo, 2023). However, the studies above, like most other existing approaches, still depend on domain expertise and inductive learning and lack the automated learning of traffic and network resource state, a common challenge with fuzzy Inference systems (Liu & Li, 2022). On the other hand, state-of-the-art DL approaches have several advantages in automating network functions compared to fuzzy systems intelligently. Thus, avoiding manually configured traffic profiles with the DL approach is a significant challenge in future intelligent networks as the human-based network dimensioning processes are expensive in cost, time, and agility. The TLM can intelligently automate the traffic profile learning process with distributed learning architecture, eliminating human intervention and making the network agile to user behavior and network changes.

The previous subsections show that the same performance can be obtained with TLM, thus eliminating the costly traffic engineering or profile-building processes. The aspects of TLM coordination and related limitations, such as the latency and detection of traffic profiles, have also been evaluated. Several experiments were conducted to show the effectiveness of the automated approach with TLM. The results have shown that it saves costs, automates traffic profile learning, and makes networks responsive to network dimensionality factors, such as traffic changes.

The TLM model evaluated in this research is an RNN-based model well-known for detecting temporal dependencies. However, spatial dependencies in the traffic profiles may also exist. Future work in this direction includes the incorporation of spatial learning models along with temporal models, such as convolutional neural networks. Since the conventional DL models lack the explainability of decisions (Wang et al., 2022), the objective function-based feature engineering to make explainable decisions requires further investigation.

Conclusions

Packet scheduling is an active area of research for mobile and fixed networks, and the researchers strive to improve performance, QoS, and network utilization. The TAOC characterization can overcome the shortcomings of traditional decentralized and independent techniques on each network node. Research has recently concentrated on such features to develop a swarm intelligence-based system. The TAOC characterization is an important area, and the manual procedures used previously are inefficient in handling traffic dynamics. The value of ML-based TAOC packet scheduling lies in its capability to bridge the gaps between network planning and functional decisions. The ML-based swarm intelligent packet scheduling framework, i.e., TLM and TIPS introduced in this article, can intelligently automate the TAOC characterization process and capture real-time traffic, network, and user dynamics. To verify this, the TIPS was ported to NS3 for performing experiments. The experimental results have shown that the proposed approach can address the challenges without affecting the network performance metrics. It has been shown in this work that with MAE from 6.38 to 8.41 (both DAL and CAL), the TAOC-based packet scheduler, along with the TLM, can maintain throughput, delay, and jitter with less than 0.05% variation as compared to the statically configured traffic profiles.

Supplemental Information

Supplemental Information 1 Code and Data

Click here for additional data file.

Additional Information and Declarations

Competing Interests

Author Contributions

Data Availability

Chan Hwang See is an Academic Editor for PeerJ.

Arif Husen conceived and designed the experiments, performed the experiments, performed the computation work, prepared figures and/or tables, authored or reviewed drafts of the article, and approved the final draft.

Muhammad Hasanain Chaudary conceived and designed the experiments, performed the experiments, prepared figures and/or tables, authored or reviewed drafts of the article, and approved the final draft.

Farooq Ahmad conceived and designed the experiments, prepared figures and/or tables, authored or reviewed drafts of the article, and approved the final draft.

Muhammad Farooq-i-Azam conceived and designed the experiments, performed the experiments, analyzed the data, performed the computation work, authored or reviewed drafts of the article, and approved the final draft.

Chan Hwang See conceived and designed the experiments, analyzed the data, performed the computation work, authored or reviewed drafts of the article, and approved the final draft.

Arfan Ghani conceived and designed the experiments, analyzed the data, performed the computation work, prepared figures and/or tables, authored or reviewed drafts of the article, and approved the final draft.

The following information was supplied regarding data availability:

TIPS Algorithm for NS3

TLM Unit Model

NS3 Trace Analyzer

Gaussian Traffic Generator

Simulation Trace Files

Experiment Code

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
