# Peer review of "Swarm intelligence-based packet scheduling for future intelligent networks"

_PeerJ Computer Science, doi:10.7717/peerj-cs.1671_

## Round 0.1 · original submission · Minor Revisions

Consider the comments provided by all reviewers, and in addition, thorough proofread is recommended.

Reviewer 4 has requested that you cite specific references. You may add them if you believe they are especially relevant. However, I do not expect you to include these citations, and if you do not include them, this will not influence my decision.

**Language Note:** The Academic Editor has identified that the English language must be improved. PeerJ can provide language editing services - please contact us at [email protected] for pricing (be sure to provide your manuscript number and title). Alternatively, you should make your own arrangements to improve the language quality and provide details in your response letter. – PeerJ Staff

·

Basic reporting

No comments

Experimental design

No comments

Validity of the findings

Valid

Additional comments

Figs. 7 to 10 appear blurred and must be made more readable.

Reviewer 2 ·

Basic reporting

The Paper investigates the current state-of-the-art methods for packet scheduling and related decision processes. Furthermore, it proposes a machine learning-based approach for packet scheduling for
agile and cost-effective networks to address various issues and challenges. The analysis of the experimental results shows that the proposed deep learning-based approach can successfully address the challenges without compromising the network performance. For example, it has been seen that with MAE from 6.38 to 8.41 with the proposed deep learning model, the packet scheduling can maintain 99.95% throughput, 99.97 % delay, and 99.94% jitter as compared to the statically configured traffic profiles.

Experimental design

It is not done in proper manner.
Proposed methodology is not enlightened in effective manner.

Validity of the findings

The Findings are not properly illustrated. And in fact analysis section is missing.

Additional comments

The Paper stands Rejected and is NOT RECOMMENDED for Publication.

Cite this review as

·

Basic reporting

This work sheds new light on the subject matter, and the findings are likely to have implications for future research. The manuscript is well-written, exhibiting clarity, coherence, and a consistent academic tone.The logical structure of the manuscript facilitates a smooth flow of information, guiding readers effectively through the research process and outcomes.

The review part can be extended and made comprehensive for understanding the relevant literature in the past.

Experimental design

The experimental design employed in the study is well-structured and demonstrates careful planning.

Validity of the findings

The use of appropriate statistical analyses and effect size measures further enhances the robustness of the results. To bolster the validity of the findings, it would be beneficial to discuss any limitations that may have affected the study's validity.

Additional comments

The discussion section is unclear and poorly organized. The discussion lacks in-depth analysis and critical interpretation of the results. Overall, the discussion section seems to be neglecting to discuss or acknowledge any negative or inconclusive outcomes, showing only better results of The efficiency of TLM in learning the TAOC characteristics. The discussion lacks recommendations for future research or potential avenues for further investigation.

Reviewer 4 ·

Basic reporting

The paper is clear and well organized.

Experimental design

The experimental setup is valid for the purpose described in the paper.

Validity of the findings

The traffic scenario is apparently reasonable. However, Figures 8 - 17 should be improved.

Additional comments

The following recent works are related to the paper:
SN Computer Science (2021) 2:473
SN Computer Science (2023) 4:345

Cite this review as

---

## Round 0.2 · accepted · Accept

Congratulations on acceptance, consider all the instructions so your manuscript can be online soon.

·

Basic reporting

The article presents a novel packet scheduling mechanism for the futuristic networks.

Experimental design

The design seems fine

Validity of the findings

Valid

Reviewer 2 ·

Basic reporting

The Revised Paper has incorporated all the comments and revisions as mentioned in the last review and now the paper can be accepted.

Experimental design

Yes, it is Ok now.

Validity of the findings

All the findings are properly revised and now the paper can be accepted.

Cite this review as

·

Basic reporting

No comment

Experimental design

No comment

Validity of the findings

No comment

Additional comments

This paper serves as an excellent introduction to the field, making it a valuable resource for researchers and practitioners seeking to harness the power of collective intelligence for solving complex problems.
The paper presents a compelling exploration of swarm intelligence-based packet scheduling for future intelligent networks. Through a well-structured analysis and practical implementation, the authors convincingly showcase the potential of swarm intelligence algorithms in optimizing packet scheduling in the context of emerging intelligent networks.

The discussion part is well written, limitations are also discussed, and the statistical work is all good. The overall quality of the paper has improved.

Reviewer 4 ·

Basic reporting

Revised manuscript

Experimental design

Revised manuscript

Validity of the findings

Revised manuscript

Additional comments

The authors have improved the manuscript according to my suggestions.

Cite this review as